# Polyphenolic Profile of *Herniaria hemistemon* Aerial Parts Extract and Assessment of Its Anti-Cryptosporidiosis in a Murine Model: In Silico Supported In Vivo Study

**DOI:** 10.3390/pharmaceutics15020415

**Published:** 2023-01-26

**Authors:** Mosad A. Ghareeb, Mansour Sobeh, Tarek Aboushousha, Marwa Esmat, Hala Sh. Mohammed, Eman S. El-Wakil

**Affiliations:** 1Medicinal Chemistry Department, Theodor Bilharz Research Institute, Kornaish El-Nile, Warrak El-Hadar, Imbaba, Giza 12411, Egypt; 2AgroBioSciences, Mohammed VI Polytechnic University, Lot 660, Hay Moulay Rachid, Ben-Guerir 43150, Morocco; 3Department of Pathology, Theodor Bilharz Research Institute, Kornaish El-Nile, Warrak El-Hadar, Imbaba, Giza 12411, Egypt; 4Department of Medical Parasitology, Faculty of Medicine, Misr University for Science and Technology, 6th October City 12566, Egypt; 5Department of Pharmacognosy and Medicinal Plants, Faculty of Pharmacy (Girls), Al-Azhar University, Cairo 11311, Egypt; 6Department of Parasitology, Theodor Bilharz Research Institute, Kornaish El-Nile, Warrak El-Hadar, Imbaba, Giza 12411, Egypt

**Keywords:** *Herniaria hemistemon* J.Gay, nitazoxanide, anticryptosporidiosis, polyphenols, LC–ESI–MS/MS, lactate dehydrogenase inhibitors

## Abstract

*Herniaria hemistemon* J.Gay is widely used in folk medicine to treat hernia. The present study aimed to annotate the phytoconstituents of *H. hemistemon* aerial-part extract and investigate its in vivo anticryptosporidial activity. The chemical characterization was achieved via the LC–ESI–MS/MS technique resulting in the annotation of 37 phytocompounds comprising flavonoids and phenolic acids. Regarding the anticryptosporidial activity, fifty dexamethasone-immunosuppressed mice were separated into five groups: GI, un-infected (normal control); GII, infected but not treated (model); GIII, infected and received NTZ, the reference drug; GIV, infected and received *H. hemistemon* extract (100 mg/kg); and GV, infected and received *H. hemistemon* extract (200 mg/kg). When GIII, GIV, and GV were compared to GII, parasitological analyses displayed highly significant differences in the mean numbers of *Cryptosporidium parvum* oocysts in the stool between the different groups. GV demonstrated the highest efficacy of 79%. Histopathological analyses displayed improvement in the small intestine and liver pathology in the treated groups (GIII, IV, and V) related to the model (GII), with GV showing the highest efficacy. Moreover, the docking-based study tentatively highlighted the potential of benzoic acid derivatives as lactate dehydrogenase inhibitors. The docked compounds showed the same binding interactions as oxamic acid, where they established H-bond interactions with ARG-109, ASN-140, ASP-168, ARG-171, and HIS-195. To sum up, *H. hemistemon* is a promising natural therapeutic agent for cryptosporidiosis.

## 1. Introduction

*Cryptosporidium* spp. are intracellular protozoan of the phylum Apicomplexa that can invade the microvillous area of epithelial cells in the digestive tract of humans and other mammals [1]. In immunocompromised patients, cryptosporidiosis can produce major complications such as severe diarrhea, dehydration, electrolyte imbalance, and hepatic and respiratory disorders [2]. Although the nature of cryptosporidiosis is widely understood, the therapeutic options are still restricted. Supportive therapy and specific treatments by Nitazoxanide (NTZ), the only Food and Drug Administration (FDA)-approved medicine in the United States, are still the only options for these serious infections. Nitazoxanide has an impact on the parasite by preventing its metabolism-necessary anaerobic energy, which makes its clearance easier [3]. Unfortunately, NTZ’s impact on children suffering from malnutrition is limited, and it has no effect on immunocompromised hosts [4].

Medicinal herbs have been widely employed in the treatment of parasitic illnesses, among them cryptosporidiosis [5,6]. These include Egyptian propolis [7], *Verbena Officinalis* [8], *Ficus carica* [9], *Olea europaea* [9], and *Asafoetida* [1]. The genus *Herniaria*, family Caryophyllaceae, comprises nearly 89 genera and 2070 species with five widespread species of this genus in Egypt, namely *H. Cyrenaica*, *H. fontanesii*, *H. hemistemon*, *H. hirsute*, and *H. arabica* [10,11]. Traditionally, *Herniaria* spp. have been utilized as antispasmodic, hypotensive, litholytic, and antidiuretic agents [12,13]. *Herniaria hemistemon* J.Gay is an herb native to North Africa and Asia, spanning from Morocco to Iran and the Arabian Peninsula but currently introduced to other parts of the world. The plant has been traditionally used in the management of hernia [12,14]. It has also shown remarkable biological activities including antioxidant [15,16], antimicrobial [16], and in vitro antischistosomal [17]. Phytochemically, different categories of compounds have been reported in the herb such as flavonoids [15], phenolic acids, and coumarins [16].

Despite its prevalent utility and numerous traditional uses, there are no adequate reports regarding the effects of *H. hemistemon* against cryptosporidiosis in a murine model. Therefore, this study aimed to annotate the chemical constituents of *H. hemistemon* aerial-part extract and to explore its activity against cryptosporidiosis in a murine model and possible modes of action via a molecular docking study.

## 2. Materials and Methods

### 2.1. Plant Material, Extraction, and Fractionation

Aerial parts (1.2 Kg) of *Herniaria hemistemon* J.Gay were collected after permission from and in compliance with relevant international guidelines and legislation of Matrouh governorate, Egypt in April 2020. The identification and authentication of the plant were performed by specialists at El-Orman Botanical Garden, Giza, Egypt. A voucher specimen was preserved at the Medicinal Chemistry Department, TBRI under the accession code (H.h.ap.2020). The dry, powdered plant materials were extracted with 85% methanol (4 × 2 L) at room temperature. The obtained extract was concentrated under reduced pressure using Rotavapor (Buchi R-300) at 40 °C. The total extract was then defatted using petroleum ether to afford 103 g (the extraction yield was 8.5%).

### 2.2. Phytochemical Analysis, Total Phenolic (TPC) and Total Flavonoid (TFC) Contents and Antioxidant Properties

The chemical components of *H. hemistemon* aerial-part extract were tentatively identified using high-performance liquid chromatography–mass spectrometry (HPLC–PDA-MS/MS). The LC system was SHIMADZU LC MS 8050 (Shimadzu, Japan, USA) coupled with a triple quadruple spectrometer with an ESI source. The separation was performed via a C18 reversed-phase column (Zorbax Eclipse XDB-C18, rapid resolution, 4.6 × 150 mm, 3.5 µm, Agilent, USA). A gradient of water and acetonitrile (ACN) (0.1% formic acid each) was applied from 5% to 30% ACN over 45 and increased to 60 over the last 15 min with a flow rate of 1 mL/min. The samples were automatically injected using autosamplerSIL-40C xs. The instrument was controlled by LC solution software (Shimadzu, Japan). The MS operated in the negative mode. TPC, TFC, DPPH (2,2-diphenyl-1-picrylhydrazyl) and total antioxidant capacity (TAC) were assayed as previously described [18].

### 2.3. In Vivo Anticryptosporidial Activity

#### 2.3.1. Animals

The experimental work was permitted by the ethical committee of Theodor Bilharz Research Institute (TBRI-REC) in accordance with internationally valid guidelines (protocol serial number: PT: 612). TBRI-REC operates in a manner consistent with the National Institute of Health (NIH) guide for the care and use of laboratory animals (Eighth edition) and adhered to the ARRIVE guidelines. Theodor Bilharz Biological Center provided laboratory male-bred, CDI-strain white Albino mice, which were about 4–6 weeks old and weighed 20–25 g. The experiments were conducted in the TBRI’s Animal House, in a well-ventilated plastic cage in conditioned rooms (24 ± 2 °C) with clean wood-chip bedding that were distant from direct sunlight, keeping a clean environment.

#### 2.3.2. Immunosuppression and Induction of Infection

A total of 50 laboratory-bred white albino male mice were orally given 0.25 µg/g/day dexamethasone sodium phosphate (Dexazone) by gavage through an esophageal tube to suppress their immune systems. Dexazone was applied daily for two weeks before oral inoculation of *Cryptosporidium* oocysts, and it was given once per week for the entire study duration for each group [19].

The mice were separated into five groups. GI: immunosuppressed and uninfected (negative control). Mice (GII–GV) were gavaged and orally infected using an esophageal tube (day 0) with *C. parvum* oocysts (concentrated from the feces of naturally infected neonatal calves and genetically identified as *C. parvum* [8]). The infection dose was approximately 1000 oocysts of *C. parvum* dissolved in 200 μL PBS for each mouse. To ensure infection establishment, fecal pellets were collected and studied after one week of mice infection (7th day post-infection (PI)).

At the 7th day PI, drugs were administered via oral gavage:-GII: Immunosuppressed and infected (model);-GIII: Immunosuppressed, infected, and received NTZ (Nanazoxid, 100 mg/5 mL suspension, Medizen Pharmaceutical industries for Utopia Pharmaceuticals) at 100 mg/kg every day for 5 successive days [20]. The doses were derived by extrapolating therapeutic human doses to animal doses [21];-GIV: Immunosuppressed, infected, and received *H. hemistemon* extract at a dose of 100 mg/kg every day for 5 days;-GV: Immunosuppressed, infected, and received *H. hemistemon* extract at a dose of 200 mg/kg every day for 5 days.

#### 2.3.3. Parasitological Examination

Fecal pellet collection was performed on the 12th day PI (the time of the end of therapeutic doses), stained using modified Zheil Nelsen stain (MZN), and examined with the x100 oil-immersion lens according to the reported procedures [22]. The parasite number was detected per gram of feces [23]. The efficacy percentages of NTZ and *H. hemistemon* were determined according to the following equation: efficacy (%) = mean value of infected non-treated group-mean value of infected treated group × 100/mean value of infected non-treated group [24].

### 2.4. Histopathological Examination

All mice were sacrificed 12 days after infection under light anesthesia by using isoflurane inhalation (Forane^®^, Baxter, UK). The jejunum, ileum, and sections from all segments of the liver tissue were collected and placed in a 10% buffered formalin solution for fixation, embedded in paraffin wax blocks, sectioned, and stained in the TBRI pathology lab with hematoxylin and eosin (H&E) to detect the pathological abnormalities [25].

### 2.5. Docking-Based Virtual Screening

The annotated compounds of *the H. hemistemon* aerial-part extract were virtually screened for their ability to inhibit lactate dehydrogenase (PDB codes: 4ND1) [26] using AutoDock Vina software [27]. Detailed methods are provided in the Appendix A.

### 2.6. Statistical Analysis

Data analysis was performed using Microsoft Excel 2016 and the statistical package for social science ‘IBM SPSS Statistics for Windows, version 26 (IBM Corp., Armonk, NY, USA)’. Continuous, normally distributed variables were represented as mean ± SE with a confidence interval of 95%. The ANOVA test was used to compare the means of normally distributed variables between groups. *p*-values ≤ 0.05 were considered statistically significant.

## 3. Results

### 3.1. Chemical Profiling

The LC–ESI–MS/MS analysis of *H. hemistemon* aerial-part extract furnished 37 secondary metabolites belonging to organic and phenolic acids, coumarins, flavonoids, and their glycosides and sulfate derivatives (Figure 1 and Table 1). Noteworthy, three coumarin derivatives were annotated in the tested extract. They furnished an [M-H]^-^
*m/z* at 339, 257, and 177, and daughter ions at 177, 121, and 133; they were assigned as esculetin glucoside, esculetin sulfate and their aglycone, esculetin, Figure 2. Several other compounds containing sulfate derivatives were also annotated in the extract. For instance, a signal furnished an [M-H]^-^
*m/z* at 247 and daughter ions at 167 ([M-H-80], typical loss of sulfate moiety) along with other fragment ions (108, 123, 155, typical daughter ions of vanillic acid); it was assigned to vanillic acid sulfate (Table 1). Another signal demonstrated an [M-H]^-^
*m/z* at 395 and daughter ions at 151, 255, 300 and 315; it was tentatively identified as isorhamnetin sulfate (Table 1).

### 3.2. Total Polyphenols, Total Flavonoids and Antioxidant Properties

The *H. hemistemon* aerial-part extract displayed considerable total phenolic (TPC) and total flavonoid (TFC) contents when assayed using the Folin–Ciocalteu and aluminum chloride methods (Table 2). As expected, it also furnished promising antioxidant potential in two assays: DPPH and TAC (Table 2). These results might be attributed to the presence of a series of polyphenolics with antioxidant properties, among them kaempferol, quercetin, isorhamnetin and myricetin, along with their glycosides and sulfate derivatives, as well as several phenolic acids (Table 1).

### 3.3. Parasitological Examination

*C. parvum* was not found in the uninfected mice (GI, the negative control group). On the other hand, all infected mice began excreting *C. parvum* oocysts, which were confirmed after seven days post-infection (PI). At the end of the experiment (12 days PI), the model (GII, infected, untreated) developed the highest oocyst intensity with a mean score of 90.4. In contrast, all treated groups showed lower oocyst intensities compared to the model (GII), with GV showing the best efficacy with an inhibition of 79% (Table 3).

### 3.4. Effects of H. hemistemon Aerial-Part Extract on the Small Intestine

Following *C. parvum* oocyst induction, several deleterious pathological changes in the small intestine were observed. These included a marked broadening of the villi with a decreased ratio of villous height to crypt length, a dense infiltration of mononuclear inflammatory cells inside the villous core, villous tip-region degeneration, and high mucin production. *Cryptosporidium* oocysts were also present along the villi brush border and in the intestinal lumen as oval-to-rounded bodies. NTZ (the reference drug) and the extract, at the two dose levels, attenuated the aforementioned effects of the extract, and at 200 mg/kg, retained a nearly normal villous pattern (Figure 3).

### 3.5. Effects of H. hemistemon Aerial-Part Extract on the Liver

Similar to the small intestine, the induction of *C. parvum* oocysts induced caused pathological changes to the liver tissue. These included hepatocyte degeneration and focal infiltration with mononuclear cells. The reference drug (NTZ) and the extract, at 100 mg/kg, moderately improved the above-mentioned effects. Interestingly, the extract, at 200 mg/kg, retained the hepatic architecture (Figure 4).

### 3.6. Docking-Based Virtual Screening

Molecular modeling has become integral in biomedical research, minimizing lab work and facilitating the discovery of the most probable molecular targets and/or signaling pathways [28]. To putatively determine the mode of action of the *H. hemistemon* aerial-part extract, we subjected the structures of all annotated compounds to molecular inverse docking experiments using the idTarget platform (http://idtarget.rcas.sinica.edu.tw accessed on 1 October 2022) [29]. We found lactate dehydrogenase (PDB ID: 4ND1) as a probable target for gallic acid, hydroxybenzoic acid, and methyl gallate with affinity scores of −9.35, −8.89, and −8.73, respectively. As a validation step, the whole annotated structures were then re-docked into the enzyme’s active site using AutoDock Vina [30]. From the docked structures, quinic acid, hydroxybenzoic acid, gallic acid, and methyl gallate relatively better scores compared co-crystalized inhibitor (oxamic acid, Table 4). They also showed several interactions such as the co-crystalized inhibitor oxamic acid, including H-bond interactions with ARG-109, ASN-140, ASP-168, ARG-171, and HIS-195 (Figure 5). This docking-based study tentatively highlighted the potential of benzoic acid derivatives as *C. parvum*-derived lactate dehydrogenase inhibitors.

## 4. Discussion

Cryptosporidiosis represents a serious health issue that can cause life-threatening diarrhea in immunocompromised people [2]. The lack of effective cryptosporidiosis therapies and vaccinations had led to the quest for an effective and safe anticryptosporidiosis therapy, particularly for immunocompromised hosts [4]. *H. hemistemon* has been proven to have antifungal effects against *Aspergillus niger* and *Candida albicans*, and antibacterial properties against both Gram-positive bacteria such as *Enterococcus faecalis, Staphylococcus aureus,* and *Bacillus subtilis* and Gram-negative bacteria such as *Escherichia coli* and *Pseudomonas aeruginosa* [16].

In this study, *Herniaria hemistemon* (*H. hemistemon*), which belongs to the family Caryophyllaceae, was tested against cryptosporidiosis in immunocompromised mice. The administration of *H. hemistemon* (GIV (100 mg/kg) and GV (200 mg/kg)) following challenged infection resulted in marked reductions in *Cryptosporidium* oocyst shedding (reduction of 61% and 79%, respectively) related to the infected control group. Interestingly, the *H. hemistemon* extract, at a dose of 200 mg/kg, displayed better activities than the reference drug, Nitazoxanide (reduction of 79% versus 67% in favor of the extract).

Additionally, a histopathological examination of the small intestine of GII (immunosuppressed and infected) revealed a deleterious effect on the structure of the intestinal mucosa compared to GI (immunosuppressed and not infected). Villous shortening and atrophy were observed, as well as a reduction in the ratio of villous height to mucosal ulceration, goblet cell depletion, crypt length, and infiltration of the lamina propria with inflammatory cells, primarily eosinophils and lymphocytes, as well as a diffuse loss of the brush border microvillous surface area. This was in line with several earlier studies [8,9]. Interestingly, these deleterious changes were inversed in GV (immunosuppressed, infected, and treated with *H. hemistemon* extract at a dose of 200 mg/kg). Noteworthily, *H. hemistemon* attenuated the adverse effects of cryptosporidiosis on the liver, where it retained a normal structure at the high dose of the extract.

The *H. hemistemon* aerial-part extract exhibited promising antioxidant activities, as well. The demonstrated activities (anticryptosporidiosis and antioxidant) could be explained by the presence of the 37 phytocompounds, detected in the extract, with reported antiparasitic and antioxidant properties. This was also supported by the docking study, which suggested that benzoic acid derivatives in the *H. hemistemon* extract (i.e., gallic acid, hydroxybenzoic acid, and methyl gallate) could act as *C. parvum* lactate dehydrogenase inhibitors and may serve as promising starting skeletons for the further development of more potent anti-*C. parvum* agents. Additionally, previous reports revealed that some identified compounds in the *H. hemistemon* extract showed potent antiparasitic effects against various types of parasites. For instance, rutin was evaluated in vitro for its ability to inhibit the replication of *C. parvum* [31]. Moreover, caffeoylquinic acids exhibited anthelmintic effects against *Entamoeba histolytica* [32]. Peña-Espinoza and his co-workers reported the antiparasitic effects of hydroxycinnamic acids, quercetin, and kaempferol derivatives against gastrointestinal parasites [33]. Furthermore, it was reported that polyphenolic compounds could be responsible for the anticryptosporidial activity of plant extracts through numerous modes of action, including interfering with essential parasite enzymes [34].

In general, the positive therapeutic effects of *Herniaria* in this study are consistent with previous reports that documented the medicinal use of the genus *Herniaria* as antispasmodic, antihypertensive, lithophytic, and diuretic agents. They demonstrated antioxidant and antimicrobial activities, as well [14]. Additionally, similar results were observed from several polyphenolic-rich extracts. These included pomegranate peels (red and white), Egyptian propolis, *Verbena Officinalis*, *Ficus carica,* and *Olea europaea* extracts; they significantly diminished the *C. parvum* oocyst count in infected mice with comparable activities to the reference drug, NTZ, and displayed a potential improvement in the shape and structure of the villi of ileal sections, as well [8,9,35].

## 5. Conclusions

The current work annotated 37 secondary metabolites from the *H. hemistemon* aerial-part extract via LC–ESI–MS/MS. It also suggested *H. hemistemon* as a promising anticryptosporidiosis agent, since it markedly decreased the *C. parvum* oocyst count in infected mice with better activities at 200 mg/kg than the reference drug, NTZ, and noticeably improved and retained the normal structure of the small intestine and liver tissue. To sum up, *H. hemistemon* could be considered as a potential candidate for further evaluation as an antiparasitic agent, food supplement, and livestock feed. Further experiments are needed to determine its toxicity and explore its individual components as well the involved mechanisms.

## Figures and Tables

**Figure 1 pharmaceutics-15-00415-f001:**
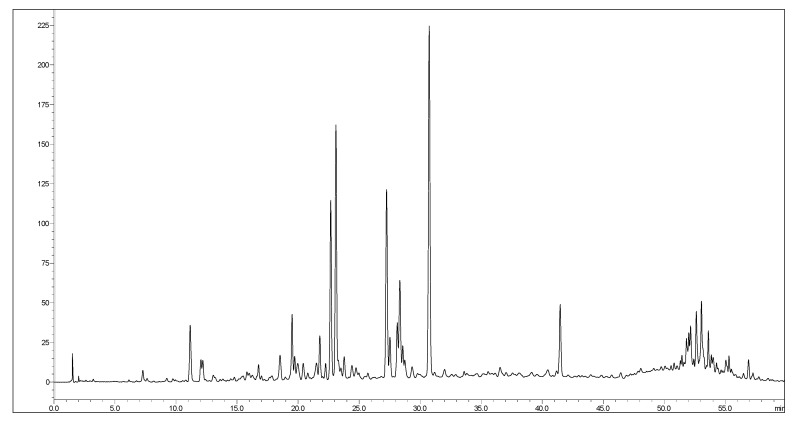
LC chromatogram of *H. hemistemon* aerial-part extract.

**Figure 2 pharmaceutics-15-00415-f002:**
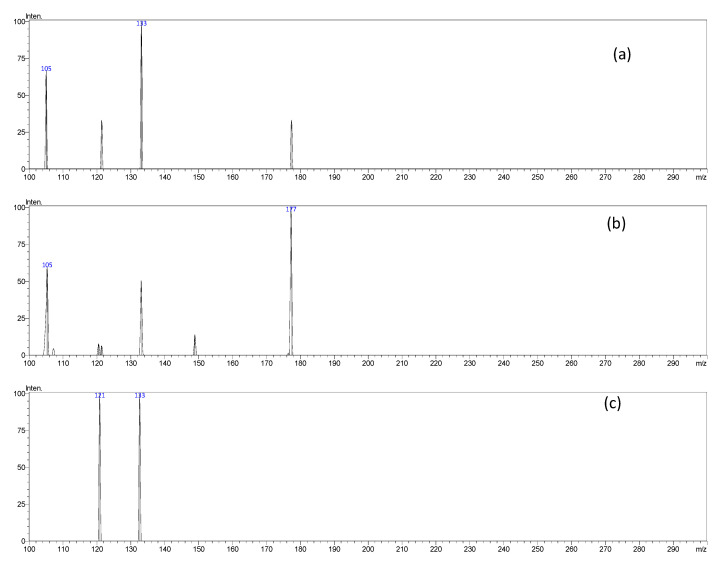
MS/MS spectra of esculetin glucoside ((**a**), compound 7), esculetin sulfate ((**b**), compound 11), and esculetin ((**c**), compound 13) from Table 1.

**Figure 3 pharmaceutics-15-00415-f003:**
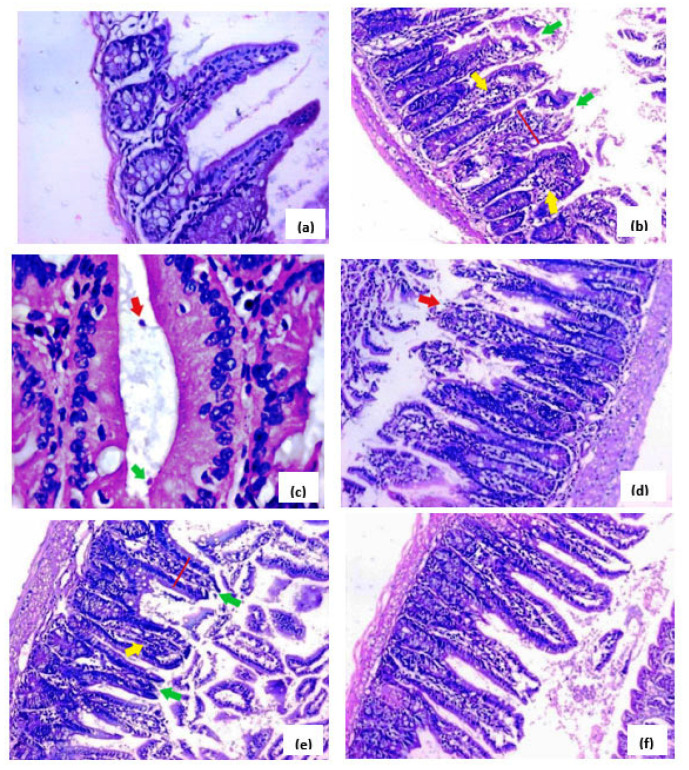
Histological photomicrographs of small intestinal sections (stained with H&E, unless mentioned otherwise, 200×). (**a**) Sections from normal mice (GI) displayed the normal structure of the villi with a preserved brush border and normal pattern of mucin secretion. (**b**) Sections from infected mice (GII) revealed a significant broadening of villi (red line), dense infiltration by mononuclear inflammatory cells (yellow arrows), and villous tip-region degeneration (green arrows). (**c**) Sections from infected mice (GII) revealed numerous adherent (green arrow) and separate *cryptosporidium* oocysts (red arrow, H&E stain, stained purple, 4–6 µm in diameter, 1000×). (**d**) Sections from infected mice (GIII) treated with the reference drug (NTZ) demonstrated mild villi broadening with focal tip-region degeneration and mild mononuclear inflammatory cell infiltration. (**e**) Sections from infected mice (GIV) treated with *H. hemistemon* extract (100 mg/kg) demonstrated moderate villous broadening (red line), infiltration by mononuclear inflammatory cells (yellow arrow), and focal degeneration of the villous tip regions (green arrow). (**f**) Sections from infected mice (GV) treated with *H. hemistemon* extract (200 mg/kg) showed a retained villous architecture, such as the recovery of an almost normal villous pattern with the occasional appearance of inflammatory cells.

**Figure 4 pharmaceutics-15-00415-f004:**
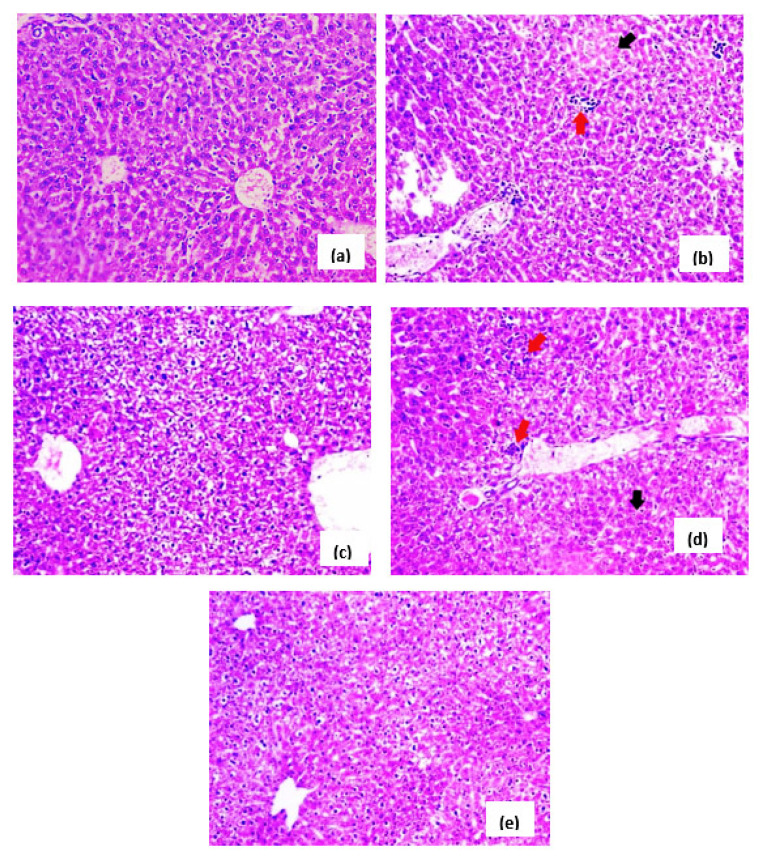
Histological photomicrographs of liver tissue (stained with H&E, unless mentioned otherwise, 200×). (**a**) Sections from normal mice (GI) showing a normal architecture of hepatic lobules. (**b**) Sections from infected mice (GII) displaying moderate degeneration of hepatocytes (black arrow) and focal infiltration with mononuclear cells (red arrow). (**c**) Sections from infected mice (GIII) treated with the reference drug, NTZ, displayed mild degeneration of hepatocytes. (**d**) Sections from infected mice (GIV) treated with *H. hemistemon* extract (100 mg/kg) showed moderate hepatocellular degeneration (black arrow) and focal infiltration by mononuclear cells (red arrow). (**e**) Sections from infected mice (GIV) treated with *H. hemistemon* extract (200 mg/kg) showed a retained hepatic lobular architecture.

**Figure 5 pharmaceutics-15-00415-f005:**
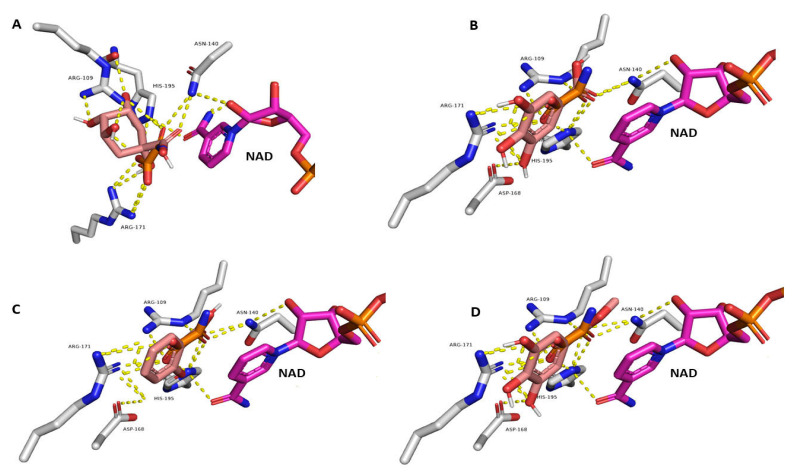
(**A**–**D**): Binding modes of quinic acid, gallic acid, hydroxybenzoic acid, and methyl gallate inside the active site of *C. parvum*-derived lactate dehydrogenase (PDB ID: 4ND1). The four structures (in brick-red color) are aligned with the structure of the co-crystalized inhibitor oxamic acid (in orange color). The structure of the co-factor nicotinamide adenine dinucleotide (NAD) is shown in pink color.

**Table 1 pharmaceutics-15-00415-t001:** Chemical constituents of *H. hemistemon* aerial-part extract.

**No.**	**Rt**	**M-H**	**MS/MS**	**Proposed Compounds**
1	1.80	191	111	Quinic acid
2	3.30	169	125	Gallic acid
3	4.86	315	153	Protocatechuic acid glucoside
4	7.20	353	191	Chlorogenic acid
5	7.59	341	135, 179	Caffeoyl glucose
6	8.05	285	153	Protocatechuic acid pentoside
7	8.67	339	177	Esculetin glucoside
8	9.01	137	108	Hydroxybenzoic acid
9	10.33	447	153, 315	Protocatechuic acid caffeoyl pentoside
10	10.31	183	125	Methyl gallate
11	10.51	257	121	Esculetin sulfate
12	11.05	353	191	Neochlorogenic acid
13	12.37	177	121, 133	Esculetin
14	12.67	329	167	Vanillic acid glucoside
15	13.75	329	167	Caffeoyl vanillic acid
16	14.59	431	153	Protocatechuic acid coumaroyl pentoside
17	15.34	337	163, 191	Coumaroylquinic acid
18	15.44	247	167	Vanillic acid sulfate
19	16.46	305	151, 287	Gallocatechin
20	18.50	337	191	Coumaroylquinic acid
21	18.62	303	137	Hydroxybenzoic acid methyl gallate
22	18.92	563	353, 383	Schaftoside
23	19.66	755	255, 301	Quercetin rhamnosyl-rutinoside
24	19.85	563	353, 383	Vicenin 1
25	19.88	625	179, 317	Myricetin rutinoside
26	21.93	739	179, 285, 575	Kaempferol dirhamnosyl-glucoside
27	22.61	769	299, 315	Isorhamnetin rhamnosyl-rutinoside
28	23.15	771	299, 315, 477	Isorhamnetin digalactosyl-pentoside
29	23.67	609	271, 301	Rutin
30	24.18	463	271, 301	Quercetin glucoside
31	25.65	593	285	Kaempferol rutinoside
32	27.12	515	161, 179, 191	Dicaffeoylquinic acid
33	27.45	623	300, 315	Isorhamnetin rutinoside
34	28.66	395	300, 315	Isorhamnetin sulfate
35	29.98	593	179, 271, 315	Isorhamnetin rhamnosyl-pentoside
36	30.64	515	161, 179, 353	Dicaffeoylquinic acid
37	32.74	409	271, 299, 314	Isorhamnetin derivative

**Table 2 pharmaceutics-15-00415-t002:** Results of TPC, TFC, DPPH and TAC of *H. hemistemon* aerial-part extract.

Sample	TPC	TFC	DPPH	TAC
	mg GAE/g Plant Extract	mg RE/g Plant Extract	IC_50_ (µg/mL)	mg AAE/g Extract
*H. hemistemon* extract	163.84 ± 3.91	61.54 ± 3.07	9.53 ± 0.67	438.67 ± 3.05
Ascorbic acid	-		3.39 ± 1.52	-

Data are presented as mean ± S.D., *n* = 3. GAE: gallic acid equivalent; RE: rutin equivalent; AAE: ascorbic acid equivalent.

**Table 3 pharmaceutics-15-00415-t003:** Results from *Cryptosporidium* spp. oocyst in stool.

Animal Groups	*Cryptosporidium* Oocysts (Mean Count/g Stool)	Inhibition %
GI (normal control)	-	-
GII (model)	90.4 ± 1.33	-
GIII (mice received the reference drug NTZ)	29.6 ± 1.35 ^a^	67%
GIV (mice received *H. hemistemon* extract (100 mg/kg)	35.7 ± 1.24 ^a,b^	61%
GV (mice received *H. hemistemon* extract (200 mg/kg)	19.1 ± 1.2 ^a,b,c^	79%

Data are expressed as mean ± SE × 10^3^, *n* = 10. ^a,b,c^ Significantly different from the model, the reference drug, and GIV at *p* ≤ 0.5, respectively.

**Table 4 pharmaceutics-15-00415-t004:** Docking scores of the annotated compounds in *H. hemistemon* aerial-part extract inside the active site of *C. parvum*-derived lactate dehydrogenase (PDB ID: 4ND1).

Compound	Docking Score
Hydroxybenzoic acid	−7.4
Quinic acid	−7.4
Gallic acid	−7.3
Methyl gallate	−7.1
Esculetin	−4.6
Dicaffeoylquinic acid	−4.4
Isorhamnetin rhamnosyl-rutinoside	−4.1
Hydroxybenzoic acid methyl gallate	−3.9
Myricetin rutinoside	−3.8
Kaempferol rutinoside	−3.6
Kaempferol dirhamnosyl-glucoside	−3.4
Isorhamnetin rutinoside	−3.3
Rutin	−3.2
Isorhamnetin sulfate	−3.2
Caffeoyl glucose	−3.1
Protocatechuic acid pentoside	−3.1
Neochlorogenic acid	−3.1
Vanillic acid glucoside	−3.1
Quercetin glucoside	−3.1
Caffeoyl vanillic acid	−3.0
Chlorogenic acid	−2.8
Vanillic acid sulfate	−2.8
Gallocatechin	−2.7
Coumaroylquinic acid	−2.6
Vicenin 1	−2.6
Protocatechuic acid glucoside	−2.5
Esculetin sulfate	−2.5
Esculetin glucoside	−2.2
Quercetin rhamnosyl-rutinoside	−2.1
Schaftoside	−1.9
Oxamic acid *	−6.1

* The reported co-crystalized inhibitor.

## Data Availability

All data generated or analyzed during this study are included in this published article and its Appendix A.

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
