# Peer review of "Polyphenolic Profile of Herniaria hemistemon Aerial Parts Extract and Assessment of Its Anti-Cryptosporidiosis in a Murine Model: In Silico Supported In Vivo Study"

_pharmaceutics, 2023, doi:10.3390/pharmaceutics15020415_

Round 1

Reviewer 1 Report

In the "Polyphenolic profile of Herniaria hemistemon aerial parts extract and assessment of its anti-cryptosporidiosis in murine model: In silico supported in vivo study" manuscript I really like the design of the study and the experimental part. This part does follow a well-established procedure and makes perfect sense, showing some potentially interesting results. Actually, I also like the computational part, as the docking has been performed using also well-established methods and since the docked compounds are small and similar to oxamic acid, there is a high chance that obtained results are accurate. One minr thing I don't like is to move the computational methods to supplementary data, as they are rather short and definitely belong to the main part of the manuscript.

There are, however, two main problems with this study. The smaller drawback is the very short Intruduction, which describes nicely the biology but fails to give some of the important information about e.g. the mode of actions of NTZ. The bigger problem is, however, this sentence: "Searching for essential protein targets for Cryptosporidium parvum in Protein Data Bank (PDB) resulted in choosing lactate dehydrogenase (PDB ID: 4ND1) as a probable target." which for me is completely not convincing and does not actually show why this particular protein was chosen as a target. There are 63 structures of proteins from Cryptosporidium parvum Iowa II in the RSCB database and authors should explain much better why this particular protein was chosen; particuralry given the fact that the mode of action of NTZ is known and it does not involve lactate dehydrogenase.

That said I find this study interesting and I'm inclined towards accepting it, though it's does provide only preliminary results. For now, however, I recommend a major revision.

Author Response

Answer to reviewers point by point

Manuscript number: pharmaceutics-2118677

Manuscript entitled: ''Polyphenolic profile of Herniaria hemistemon aerial parts extract and assessment of its anti-cryptosporidiosis in murine model: In silico supported in vivo study''

By: Mosad A. Ghareeb, Mansour Sobeh, Tarek Aboushousha, Marwa Esmat, Hala Sh. Mohammed, Eman S. El-Wakil

-------------------------------------------------------------------------------------------------------

Dear Respected Editor and Reviewers,

First, we would like to thank all reviewers for devoting their valuable time for reviewing rigorously our manuscript. All comments/suggestions are valid and useful to improve the quality of our manuscript.

Please find enclosed our rebuttal replay for reviewers’ comments/suggestions point by point. All modifications in the revised file are marked in red colour.

-------------------------------------------------------------------------------------------------------

Reviewer #1:

1) In the "Polyphenolic profile of Herniaria hemistemon aerial parts extract and assessment of its anti-cryptosporidiosis in murine model: In silico supported in vivo study" manuscript I really like the design of the study and the experimental part. This part does follow a well-established procedure and makes perfect sense, showing some potentially interesting results. Actually, I also like the computational part, as the docking has been performed using also well-established methods and since the docked compounds are small and similar to oxamic acid, there is a high chance that obtained results are accurate. One minor thing I don't like is to move the computational methods to supplementary data, as they are rather short and definitely belong to the main part of the manuscript.

# We are trying to keep our manuscript concise; therefore, we kept at the suppl. file. Based on your wise recommendation, this part has been moved to the main file:

2.5. Docking-based virtual screening

The annotated compound of H. hemistemon aerial parts extract were virtually screened for their abilities to inhibit lactate dehydrogenase (PDB codes: 4ND1) [26] using AutoDock Vina software [27]. Detailed methods are provided at the supplementary file.

2) There are, however, two main problems with this study. The smaller drawback is the very short Introduction, which describes nicely the biology but fails to give some of the important information about e.g. the mode of actions of NTZ.

# We added the mode of action to the introduction section.

3) The bigger problem is, however, this sentence: "Searching for essential protein targets for Cryptosporidium parvum in Protein Data Bank (PDB) resulted in choosing lactate dehydrogenase (PDB ID: 4ND1) as a probable target." which for me is completely not convincing and does not actually show why this particular protein was chosen as a target. There are 63 structures of proteins from Cryptosporidium parvum Iowa II in the RSCB database and authors should explain much better why this particular protein was chosen; particularly given the fact that the mode of action of NTZ is known and it does not involve lactate dehydrogenase. That said I find this study interesting and I'm inclined towards accepting it, though it's does provide only preliminary results. For now, however, I recommend a major revision.

# In order to putatively find out the probable protein target for the annotated compounds in H. hemistemon aerial parts extract, we subjected their structures to molecular inverse docking experiments using the idTarget platform (http://idtarget.rcas.sinica.edu.tw). This platform can dock the query structures against almost all proteins hosted in Protein Data Bank (PBD), and then retrieve the results as a list of binding affinity scores. Accordingly, we found lactate dehydrogenase (PDB ID: 4ND1) as a probable target for gallic acid, hydroxy benzoic acid, and methyl gallate with affinity scores of -9.35, -8.89, and -8.73, respectively. As a validation step, the whole annotated structures were then re-docked into the enzyme’s active site using AutoDock Vina.

We hope that we have satisfactorily handled all the concerns highlighted. Thank you so much for everything and I hope to accept my manuscript in this form.

Best regards,

Associate Professor/ Mosad Ahmed Ghareeb

Department of Medicinal Chemistry, Theodor Bilharz Research Institute, Kornaish El Nile, Warrak El-Hadar, Imbaba (P.O. 30), Giza 12411, Egypt

Tel.: 00201012346834

Reviewer 2 Report

Cryptosporidium spp. represents a parasite with pathological implications in numerous species, thus the medical interest in the genus was constantly increasing especially in the last decades (S. Tzipori, H. Ward / Microbes and Infection 4 (2002) 1047–1058). The aggressiveness of the parasite as well as the difficulties in treating the disease rely on its capacity to persist in the host due to  repeated first-generation merogony and the production of sporulated thin-walled oocysts, a characteristic quite distinct from other coccidia. Thus, all attempts to detect alternative therapies are welcome.

The authors follow in this paper a logical path based on the chemical composition reflected in the in vivo anti-cryptosporidium effect, relying on the influence of the plant extract on the structure of the gut and liver and supported at molecular level by the docking technique results.

Nevertheless, improvement can be brought to a better use of the results and a higher impact of the research.

More detailed information on the biological effects on protozoal parasites of plant extracts/Herniaria spp. could be useful for the reader to correctly place the findings of the presented research in an "importance" framework.

The authors could better explain the importance of docking and its biological significance in what concerns the extract of Herniaria hemistemon

Please see other comments in the attached document.

Author Response

Answer to reviewers point by point

Manuscript number: pharmaceutics-2118677

Manuscript entitled: ''Polyphenolic profile of Herniaria hemistemon aerial parts extract and assessment of its anti-cryptosporidiosis in murine model: In silico supported in vivo study''

By: Mosad A. Ghareeb, Mansour Sobeh, Tarek Aboushousha, Marwa Esmat, Hala Sh. Mohammed, Eman S. El-Wakil

-------------------------------------------------------------------------------------------------------

Dear Respected Editor and Reviewers,

First, we would like to thank all reviewers for devoting their valuable time for reviewing rigorously our manuscript. All comments/suggestions are valid and useful to improve the quality of our manuscript.

Please find enclosed our rebuttal replay for reviewers’ comments/suggestions point by point. All modifications in the revised file are marked in red colour.

-------------------------------------------------------------------------------------------------------

Reviewer #2:

1) Cryptosporidium spp. represents a parasite with pathological implications in numerous species, thus the medical interest in the genus was constantly increasing especially in the last decades (S. Tzipori, H. Ward / Microbes and Infection 4 (2002) 1047–1058). The aggressiveness of the parasite as well as the difficulties in treating the disease rely on its capacity to persist in the host due to repeated first-generation merogony and the production of sporulated thin-walled oocysts, a characteristic quite distinct from other coccidia. Thus, all attempts to detect alternative therapies are welcome. The authors follow in this paper a logical path based on the chemical composition reflected in the in vivo anti-cryptosporidium effect, relying on the influence of the plant extract on the structure of the gut and liver and supported at molecular level by the docking technique results. Nevertheless, improvement can be brought to a better use of the results and a higher impact of the research. More detailed information on the biological effects on protozoal parasites of plant extracts/Herniaria spp. could be useful for the reader to correctly place the findings of the presented research in an "importance" framework.

# We improved it.

2) The authors could better explain the importance of docking and its biological significance in what concerns the extract of Herniaria hemistemon

# We added a brief explanation about the importance of docking in general. In addition, we commented on our docking results revealing its impact on our findings.

3)  Line 21 “Hernia” should not be capitalized.

# Done.

4) Line 31: “GV demonstrated the most efficacy with a 79%.” Maybe “GV demonstrated the highest efficacy with a 79%. inhibition”????

# Done.

5) Lines 31-32: “The villous broadening and inflammatory infiltrates in the small intestine improved to varying degrees” should be rephrased.

# Done.

6)  Lines 53-54 should be moved to the beginning of paragraph 2 in the introduction.

# Done.

7) Lines 66-67: Only the plant’s use in a murine model is absent or in other cases of cryptosporidiosis as well?.

# There are no studies that document the use of the plant Herniaria in cryptosporidiosis, but the plant Herniaria was used in vivo in Europe as a treatment for hernias (Sakkir et al., 2012).

Sakkir, S., Kabshawi, M., and Mehairbi, M. (2012). Medicinal plants diversity and their conservation status in the United Arab Emirates (UAE). Journal of Medicinal Plants Research, 6(7), 1304-1322.

8) Line 130: “The efficacy percentage of each drug was determined”. Are the plant extracts considered drugs in this case?

# We corrected this in the manuscript and write the names NTZ and H. hemistemon.

9) Lines 136-137: “Small intestinal and liver tissues were”. Could the authors be more specific about the portion, amount, etc.?.

# We took the jejunum and ileum parts of the small intestine and we took sections from all segments of the liver tissue. We added this to the manuscript.

10) Lines 185-186: “It is worth noting that the extract, at 200 mg/kg dose, furnished 185 better activities than the reference drug, NTZ, Table 3.” – this is a discussion of the results.

# We rephrased this in the manuscript.

11) Lines 196-198: “Intersestingly, the reference drug and the extract, at the 196 two dose levels, attenuated the afromentioned effects. Noteworthy, the extract, at (200 197 mg/kg), furnished nearly normal villous pattern, Figure 3.” - These are discussions and please, correct the wrong words.

# Corrected.

12) Lines 209-216: The wording "improvement in the pathological changes" or "improved pathology abnormalities" might not be the most appropriate to indicate a post-therapy improvement of the gut condition (should be reconsidered). Thus, expressions such as "(e) Sections from infected mice (G IV) treated with H. hemistemon extract (200 mg/kg) showing retained hepatic lobular architecture.", in Fig. 4 are more appropriate.

# We rephrased this in the manuscript.

13)  Line 240: “relatively low sores” must be “relatively low scores”.

# Corrected.

14) Line 268: “immunocompromised murine model” should be rephrased, not the model is immunocompromised but the mice used in the model.

# Done.

15) Line 273: “reduction 61% and 79%” should be corrected “reduction of 61% and 79%”.

# Done.

16) Line 276: “to the favor of H. hemistemon extract” should be “in favour of …”.

# Corrected.

17) Lines 277-280: The following paragraph should be placed before discussing the anti-cryptosporidial effects of the plant, as a general mention on the biological effects of the plant: “Herniaria hemistemon was proved to have antifungal effect; it had an inhibitory activity on Aspergillus niger and candida albicans [13]. Moreover, it showed significant effect against both gram positive bacteria as Staphylococcus aureus, Enterococcus faecalis, and Bacillus subtilus (Bacillus subtilis!) as well as gram negative bacteria as Pseudomonas aeruginosa and Escherichia 280 coli [13].”

# Moved.

18) Lines 315-319: The authors should more explicitly discuss this paragraph: “On the other hand, docking-based study suggested that benzoic acid derivatives within the H. hemistemon extract (i.e., gallic acid, hydroxybenzoic acid, and methyl gallate) might function as C. parvum lactate dehydrogenase inhibitors. The presented findings revealed that the small scaffold of these metabolites align perfectly with that of oxamic acid (i.e., the reported co-crystalized inhibitor), and hence, they might serve as a promising starting point for the future development of more potent anti- C. parvum agents.”

# Further explanation about how we suggested C. parvum lactate dehydrogenase as a probable target for the aforementioned compounds was added accordingly in the revised manuscript. Additionally, this paragraph was rephrased accordingly as required in the revised manuscript.

We hope that we have satisfactorily handled all the concerns highlighted. Thank you so much for everything and I hope to accept my manuscript in this form.

Best regards,

Associate Professor/ Mosad Ahmed Ghareeb

Department of Medicinal Chemistry, Theodor Bilharz Research Institute, Kornaish El Nile, Warrak El-Hadar, Imbaba (P.O. 30), Giza 12411, Egypt

Tel.: 00201012346834

Round 2

Reviewer 1 Report

The improved manuscript "Polyphenolic profile of Herniaria hemistemon aerial parts extract and assessment of its anti-cryptosporidiosis in murine model: In silico supported in vivo study" is, in my opinion, ready to be published in Pharmaceutics.